# Spatial Variation and Temporal Instability in the Growth/Climate Relationship of White Birch (*Betula platyphylla* Suk) in the Changbai Mountain, China

**Yangao Jiang [1], Yuting Cao [1], Shijie Han [2,\*], Junhui Zhang [3,\*] and Lin Hao [4]**

[1] Experimental Teaching Center, Shenyang Normal University, Shenyang 110034, China; jiangyangao-jyg@163.com (J.Y.); caoyuting4640@163.com (Y.C.)
[2] School of Life Sciences, Henan University, Kaifeng 475004, China
[3] Institute of Applied Ecology, Chinese Academy of Sciences, Shenyang 110016, China
[4] College of Life Sciences, Shenyang Normal University, Shenyang 110034, China; haolinwj2001@163.com
\* Correspondence: hansj@iae.ac.cn (S.H.); jhzhang@iae.ac.cn (J.Z.); Tel.: +86-24-8397-0443 (J.Z.)

**Abstract:** Tree growth in mountain ecosystems is affected by complex environments, and its relationship with climatic and environmental factors varying with elevation. In order to examine the spatial variation and temporal stability of the growth/climate relationship of *Betula platyphylla* (*BP*), the dendrochronological method was used to analyze the radial growth/climate relationship between 1946 and 2016 of the *BP* trees along an altitudinal gradient in the Changbai Mountain of northeast China. Our results showed that the mean sensitivity of *BP* was higher than that of other species studied in Changbai Mountain. The growth/climate relationship of *BP* trees varies with altitude, and this conclusion has reached a consensus from the study of tree growth response to climate change. More specifically, at low altitudes (550–995 m a.s.l.), the radial growth of *BP* is mainly affected by spring precipitation and temperature in May and October of the current year. However, at high-altitude areas (1210–1425 m a.s.l.), it is mainly affected by the temperature in September of the previous year and May of the current year. Furthermore, the growth/climate relationship of *BP* trees showed temporal instability. After 1970, the rise in temperature inhibited the growth of *BP* at low altitudes and promoted the growth of *BP* trees at high altitudes. In the context of continued warming in the future, the white birch stands in Changbai Mountain will move to higher altitudes.

**Keywords:** *Betula platyphylla*; radial growth; temporal instability; climate variability; Changbai Mountain

## 1. Introduction

The rising global temperature has a major influence on high-altitude mountain ecosystems of the continental interior of the mid-high latitudes in the Northern Hemisphere [1], causing significant changes in species phenology, biomass, diversity, distribution, and ecosystem processes in this region [2–16]. There is increasing evidence that since the 1970s, the northeast China region has experienced rapid warming, leading to a warming and drying trend in this region [17–20]. The Northeast Forest is one of the three major state-owned forest areas in China, which is extremely sensitive to climatic and environmental changes [21–24], and the relationship of its dynamics with climatic factors has been receiving more and more attention.

The increasing temperature is affecting the growth of trees in Northeast China [22,25]. The results from models indicated that the distribution of dominant conifer species in northeast China will tend to move northward and to higher altitudes, and the forest structure and composition will change under the coming climate warming and drying scenarios [25–27]. Meanwhile, based on empirical/observational evidences, dendrochronological studies by Jiang et al. (2016) [8] suggested that the radial growth of *Larix gmelinii* was decreasing under the observed temperature increases in the Greater Khingan Mountains of northeast

China, while the radial growth will benefit in the permafrost zone of this area due to the increment of the thaw depth of the permafrost [9]. Forest microclimate affects the physiological processes of trees and then controls growth [28]. Correlation analyses indicated that the response characteristics of trees to climatic factors in this area were likely to be different due to latitude [8,29] and altitude variations [30–37]. At present, the vast majority of growth/climate-response analyses focus on conifer trees, while using deciduous broad-leaved species to study the climate/growth relationship are still rare, resulting in a lack of a comprehensive understanding of the dynamics of forests under climate warming conditions in northeast China.

White birch (*Betula platyphylla* Suk., abbreviated as *BP*), also known as Japanese white birch, Asian white birch, Siberian silver birch, or Manchurian birch, can grow well under different environmental conditions due to its drought resistance and frost resistance [38], maintaining a continuous forest cover in the forests of east Asia. It has an economic value, can reach up to 27 m in height, and can survive for 120–140 years [39] within the light taiga habitat [40]. In the past years, several studies of white birch have been carried out in other countries outside of China, such as Mongolia [39], Russia [40,41], and Japan [42]. However, these studies revealed that the *BP* trees showed inconsistent responsiveness to climate factors, and site-specific growth responses to climate were observed [39,41,42]. In northeastern China, *BP* is often the main species for mixed stands of *Larix gmelinii* Rupr., *Populus davidiana* Dode, *Pinus koraiensis* Sieb., and *Quercus mongolica* Fisch., or they can exist as a pure forest. It is also one of the pioneer species of forests in northeastern China, which dominates after disturbances, such as clear-cutting or fires, thanks to its germination capacity [43]. So far, the research on the response of the growth of *BP* trees to climatic factors in northeastern China has not been carried out. Thus, the relationship between the growth of white birch and climatic factors, and the impacts of climate warming on its growth, remain unknown.

Changbai Mountain (CBM) is the core area of forests in northeast China, and the growth of trees within it is very sensitive to climate change [32–35,44]. It is one of the best protected areas of forest vegetation in China. Since the establishment of the Qing Dynasty (AD 1636), the mountain closure policy was implemented in CBM, because the emperors considered it as the birthplace of the Dynasty. In the 1980s, CBM was designated as a nature reserve. So, it has never been widely affected by human activities. Meanwhile, it covers a large area of pure white birch (*BP*) forests, and the *BP* forests are distributed at an altitude of 550–1425 m a.s.l. Thus, CBM is an ideal experimental platform from where to study the climate/growth relationships for *BP*.

In this paper, a dendroecological approach was used to explore the spatio-temporal variation in the responses of *BP* forest growth to climate change. We mainly want to solve three problems: (1) exploring the most important climatic factors affecting the growth of *BP* in Changbai Mountain; (2) exploring the extent of how *BP* react to climate variability; and (3) exploring whether the climate/growth relationships are consistent or non-stationary over time. This study is the first time the relationships between climatic factors and growth of white birch trees in the forests of northeast China are explored. Through this paper, we can better understand the forests dynamics in northeast China in the context of climate change. In addition, this study can also provide data to support policy-making for forest protection and management and as well as for forest succession models.

## 2. Materials and Methods

### 2.1. Study Area and Sample Collection

The study area is located at the Changbai Mountain (CBM) Natural Reserve in northeast China (41°31′–42°28′ N, 127°9′–128°55′ E) (Figure 1), where the climate is affected by the temperate continental monsoon, which is characterized by moist summers and cold, windy winters [44]. The annual average temperature (Ta) is 2.14 ($\pm$0.75) °C, and the annual precipitation (P) is 833 ($\pm$114) mm from 1958 to 2016 (Figure 2). About 88.4% of annual precipitation occurs from April to September. The sampling was conducted in

November 2016. Five sampling sites were chosen to represent the elevation range of *BP*: the low-elevation sites (E550 at 550 m a.s.l.; E760 at 760 m a.s.l.; and E995 at 995 m a.s.l.), and the high elevation sites (E1210 at 1210 m a.s.l.; and E1425 at 1425 m a.s.l.).

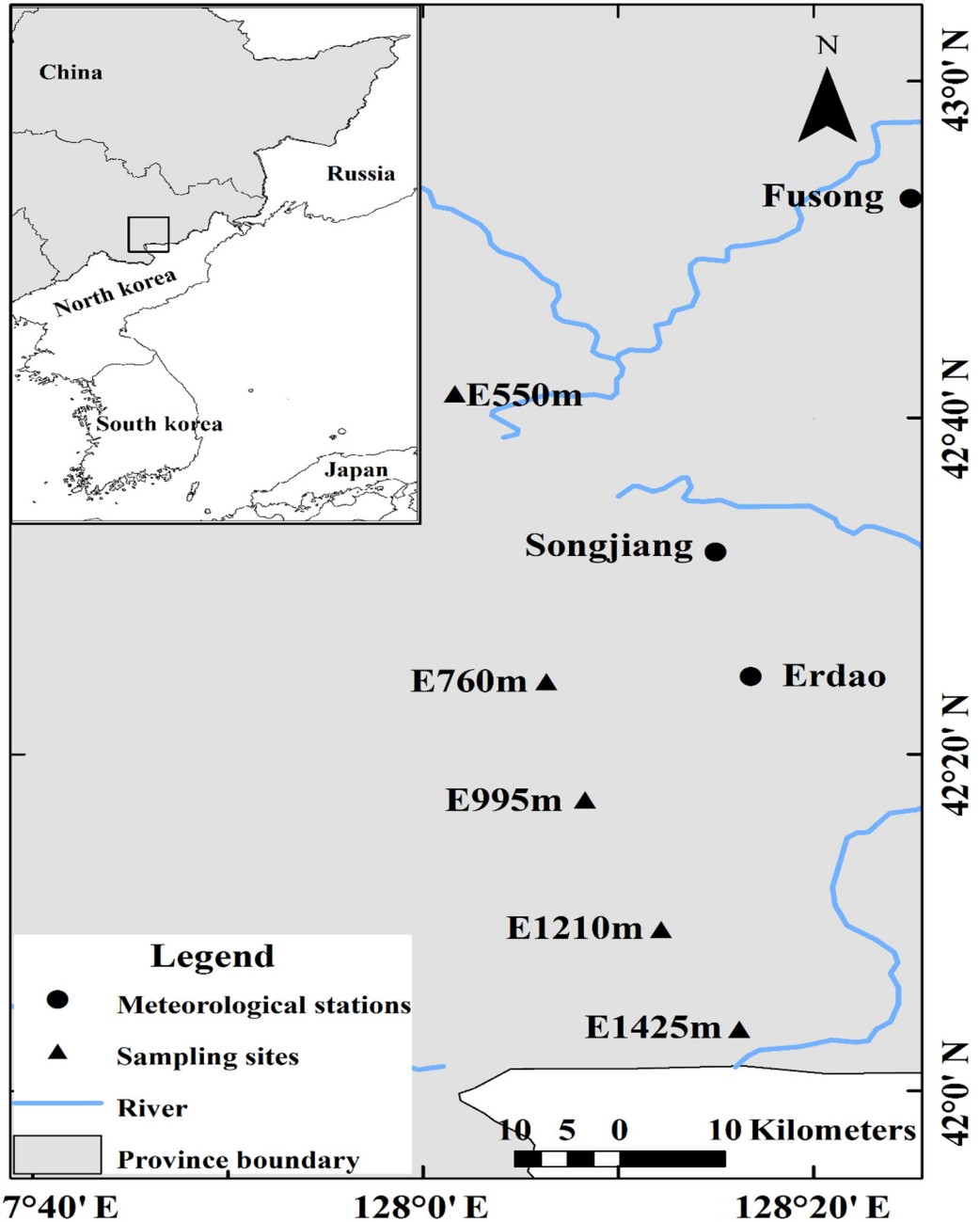

**Figure 1.** The sampling sites and weather stations in the Changbai Mountain, China. The black triangles are sampling points, and the black dots are weather stations.

The sampling sites are all on the north slope and are covered by similar weather patterns. The P is from 767 mm at E550 to 886 mm at E1425; the Ta is from 3.8 °C at E550 to 0.4 °C at E1425 m. The low-altitude sampling point was located at the lowest edge of the species' distribution, while the high-altitude sampling point was located on the upper tree line. The vertical distance between the lowest edge and upper tree line was about 925 m. All the selected forests were natural secondary forests, which can be regarded as pure birch forests, because there were only some pine seedlings. In order to minimize the impact of

non-climatic factors on the growth of trees, the selected *BP* forests were natural and open forests with no signs of recent fires or human disturbances. According to the prevalence of white birch, two or three plots of 30 m × 30 m were established for each sampling site. All of the healthy, presumably oldest and largest canopy trees (with a height of 20–27 m and a diameter of 19–32 cm) were chosen and then one or two oppositely oriented increment cores were sampled at the trunk 1.3 m above the ground. In some cases, when the core was partially fragmented or rotted, only one core was taken. At each site, 22–33 trees were selected; thus, in total, 272 cores from 142 *BP* living trees were collected. The slope of the sampling points was from 0° to 15°.

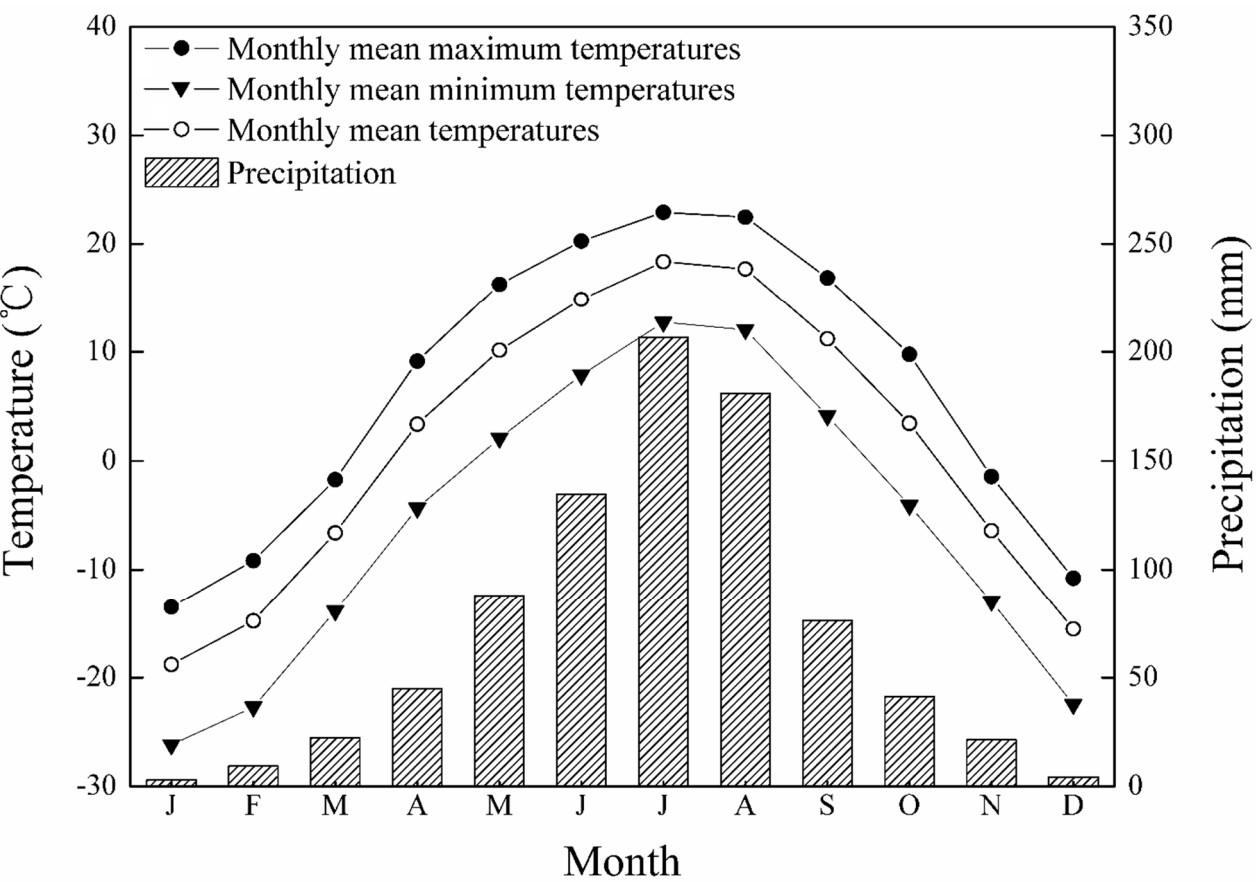

**Figure 2.** Total precipitation (in mm) and mean monthly temperature (in °C) in Changbai Mountain (AD 1946—2016) based on the CRU data.

### 2.2. Development of Ring-Width Chronologies

In the laboratory, the samples were pretreated, naturally air dried, glued to wooden holders, and then sanded with successively finer grits of sandpaper to highlight the tree rings [45,46]. The tree rings were cross-dated under a binocular microscope, and then the annual ring width was measured using a Velmex measuring system interfaced with the 'Time Series Analysis Program' (TSAP; Frank Rinntech, Heidelberg, Germany) with an accuracy of 0.001 mm. The COFECHA program was used in the quality control of the cross-dating and the measurements [45]. To eliminate the effects of stand dynamics, age, and any other non-climate-related growth variation, the cross-dated tree-ring data were detrended using two different techniques—negative exponential (EXP) and regional curve standardization (RCS), via ARSTAN [46]. Each technique produced three chronologies in the program ARSTAN: autoregressive (ARS), residual (RES), and standard (STD).

The STD chronologies in both techniques contain the most climate signals, while the STD chronology with an EXP detrending was the most suitable chronology because it

contained more low frequency signals (Figure 3). Analyses were restricted to the period with an expressed population signal (EPS) greater than 0.85 [46,47]. Thus, according to Table 1, site E760 has an EPS value greater than 0.85 from 1951, while the other four sites have EPS values greater than 0.85 from 1946. The period from 1946 to 2016 was used for further analysis. In addition, in order to test if the *BP* trees were sensitive to high-frequencies climate variations, correlations between the RES chronologies and climate data were also carried out.

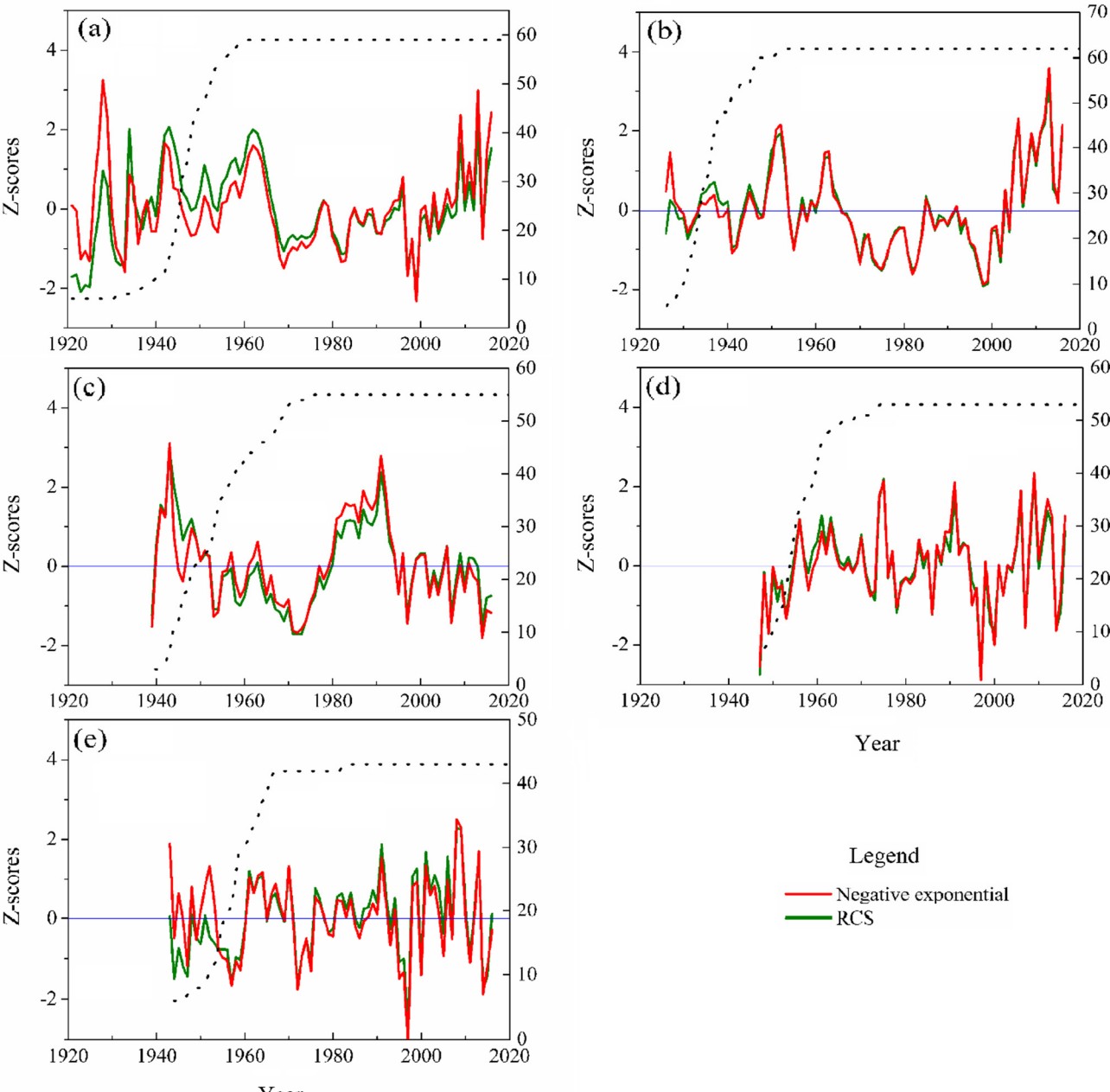

**Figure 3.** The EXP (negative exponential, red) and RCS (regional curve standardization, green) chronologies and numbers of cores from AD 1921–2016 (1425 m a.s.l.) (**a**); 1926–2016 (1210 m a.s.l.) (**b**); 1939–2016 (995 m a.s.l.) (**c**); 1946–2016 (760 m a.s.l.) (**d**); and 1943–2016 (550 m a.s.l.) (**e**).

**Table 1.** Sampling sites and statistical characteristics of white birch standard chronologies along an altitudinal gradient from 550 to 1425 m a.s.l. in Changbai Mountain, China.

| Parameters | E550 | E760 | E995 | E1210 | E1425 |
|---|---|---|---|---|---|
| Elevation | 550 | 760 | 995 | 1210 | 1425 |
| Latitude | 42°41′ | 42°24′ | 42°15′ | 42°09′ | 42°03′ |
| Longitude | 128°01′ | 128°06′ | 128°09′ | 128°12′ | 128°16′ |
| No. of trees/cores | 22/43 | 28/53 | 28/55 | 33/62 | 31/59 |
| Period | 1943–2016 | 1947–2016 | 1939–2016 | 1926–2016 | 1921–2016 |
| Mean sensitivity (MS) | 0.277 | 0.29 | 0.295 | 0.304 | 0.309 |
| Standard deviation (SD) | 0.18 | 0.21 | 0.22 | 0.14 | 0.19 |
| Skewness | 1.27 | 1.157 | 0.345 | 1.005 | 1.021 |
| Kurtosis | 4.36 | 4.168 | 2.322 | 3.693 | 4.231 |
| Signal-to-noise ratio (SNR) | 8.5 | 11.6 | 11.7 | 9.3 | 8.7 |
| Mean ring width (mm) | 1.58 ± 0.06 | 1.16 ± 0.05 | 1.42 ± 0.07 | 1.03 ± 0.03 | 1.13 ± 0.06 |
| Mean inter-series correlation (RBAR) | 0.566 | 0.559 | 0.613 | 0.545 | 0.587 |
| Auto correlation | 0.582 | 0.713 | 0.708 | 0.784 | 0.703 |
| Expressed population signal (EPS) | 0.92 | 0.92 | 0.95 | 0.93 | 0.96 |
| First year where EPS > 0.85 | 1946 | 1951 | 1945 | 1929 | 1927 |

*2.3. Climate Data and Climate–Growth Relationships*

Climate data (precipitation and temperature, 1946–2016) were obtained from the CRU TS 4.02 0.5°× 0.5°grid box (Climate Research Unit, UK) at 128–129° N, 42–43° E (http://www.cru.uea.ac.uk/) [48]. The Palmer Drought Severity Index (PDSI) is the parameter that was applied to test the severity of the drought. PDSI data for 1946–2016 were obtained from the 0.5° × 0.5° CRU scPDSI 3.26 early grid box at 42–43° N, 128–129° E (http://climexp.knmi.nl). To test the reliability of the gridded meteorological data, we compared it with the measured meteorological data of three meteorological stations, including Songjiang (128°15′ E, 42°32′ N), Fusong (127°34′ E, 42°06′ N), and Erdao Station (128°25′ E, 42°53′ N), in Changbai Mountain area. The meteorological data provided by these three weather stations are all after 1958. There is a significant correlation between the grid data and monthly climate variables at the three stations from 1958 and 2016 ($r = 0.90$, $p < 0.001$). An upward or downward trend was detected in the climate data (Figure 3) and was removed through linear regression; after that, the Pearson correlations between the detrended climate data and STD/RES chronologies between 1946 and 2016 were analyzed. An analysis of the climate dynamics from 1946 and 2016 showed that, since 1970, the monthly mean temperatures have risen rapidly, and the severity of drought has gradually increased (Figure 4). To verify whether warming has an effect on the relationship between tree growth and climatic factors, correlations between the radial growth and climate factors before and after 1970 were conducted.

Correlations between the tree-ring chronologies and monthly climatic variables from the previous year's September (the mean temperature > 5 °C) to the current year's October (the mean temperature > 0 °C) were carried out, and five climate variables were applied for the analyses, including monthly maximum temperature (Tmax), monthly minimum temperature (Tmin), monthly mean temperature (T), monthly total precipitation (Pre), and monthly Palmer Drought Severity Index (PDSI). In addition, in order to explore if the growth of *BP* could be linked to seasonal climate conditions, the correlations between seasonal climate data and radial growth was also calculated. According to the T and the Tmin of the study region, four seasons were delineated: (a) the winter dormant period (WD) (T < 0 °C), during the previous year's November and current year's March; (b) the

early growing season (BG) (Tmin < 5 °C), during April and May; (c) the growing season (GS), from June to August; and (d) the end of the growing season (EG) (Tmin < 5 °C), from the current year September to current year October.

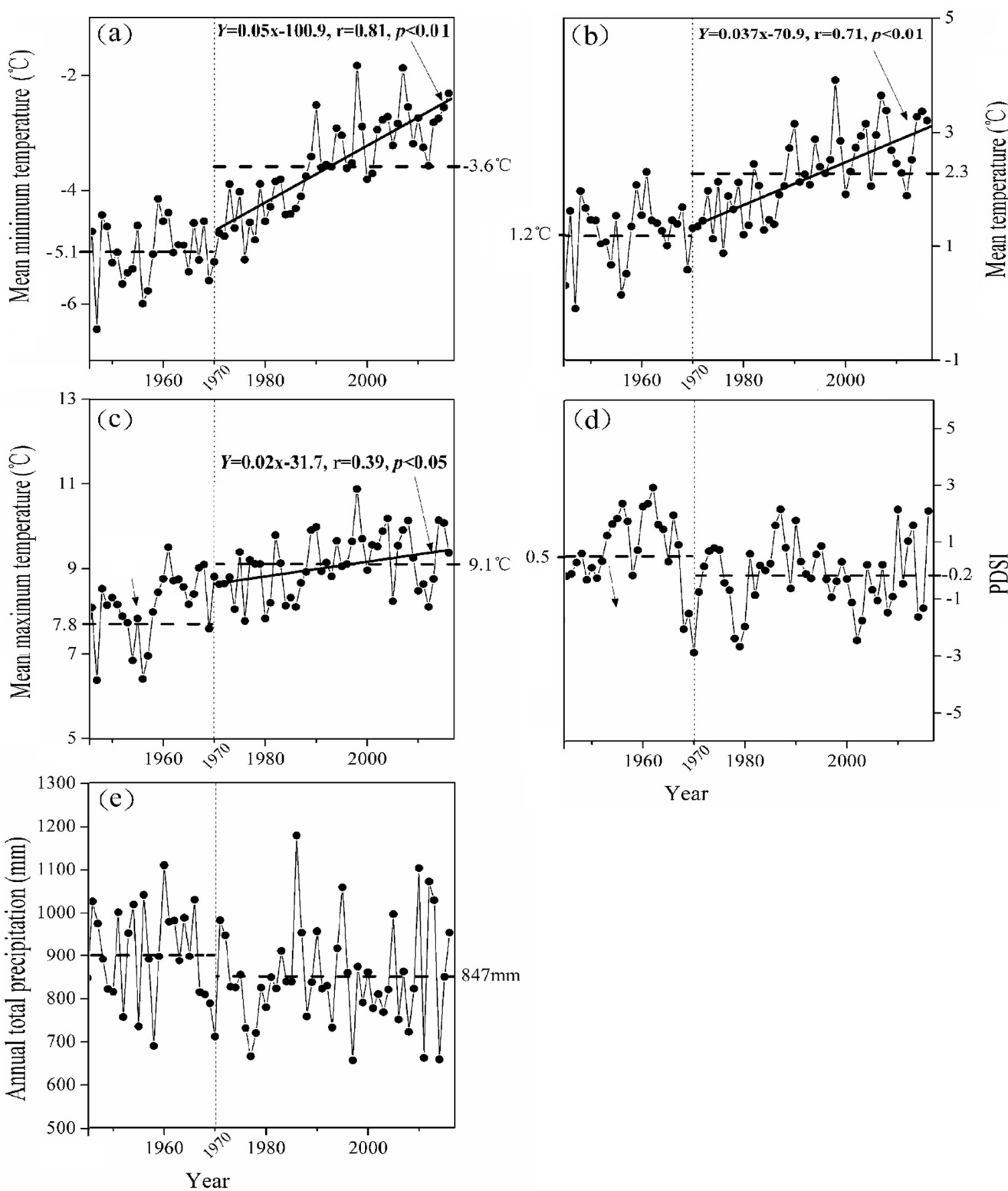

**Figure 4.** Variation in the annual (**a**) mean minimum temperature, (**b**) mean temperature, (**c**) mean maximum temperature, and (**d**) PDSI between 1946 and 2016, (**e**) annual total precipitation from the gridded data.

## 3. Results

### 3.1. Standard Chronologies' Characteristics

The five final standard chronologies with an EPS greater than 0.85 between 550 m a.s.l. and 1425 m a.s.l. is displayed in Figure 4. The time spans of the five *BP* chronologies were 1946–2016 (E550), 1951–2016 (E760), 1945–2016 (E995), 1929–2016 (E1210), and 1946–2016 (E1425), respectively (Table 1). The mean ring width ranged between 1.03 and 1.58 mm, which showed a significant negative correlation with elevation (Figure 5a). RBAR is the mean inter-series correlation [45], ranging from 0.545 to 0.613. The standard deviation (SD) of the tree-ring widths in the five sites ranged from 0.14 to 0.22, whereas the mean sensitivity (MS), describing the interannual variation in the ring widths, ranged from 0.281 to 0.377 (Table 1). Significant first-order autocorrelations (AC1) were showed in all tree-ring series (Table 1), ranging from 0.482 to 0.756, indicating a significant correlation between the current-year ring width and previous year's ring width [28].

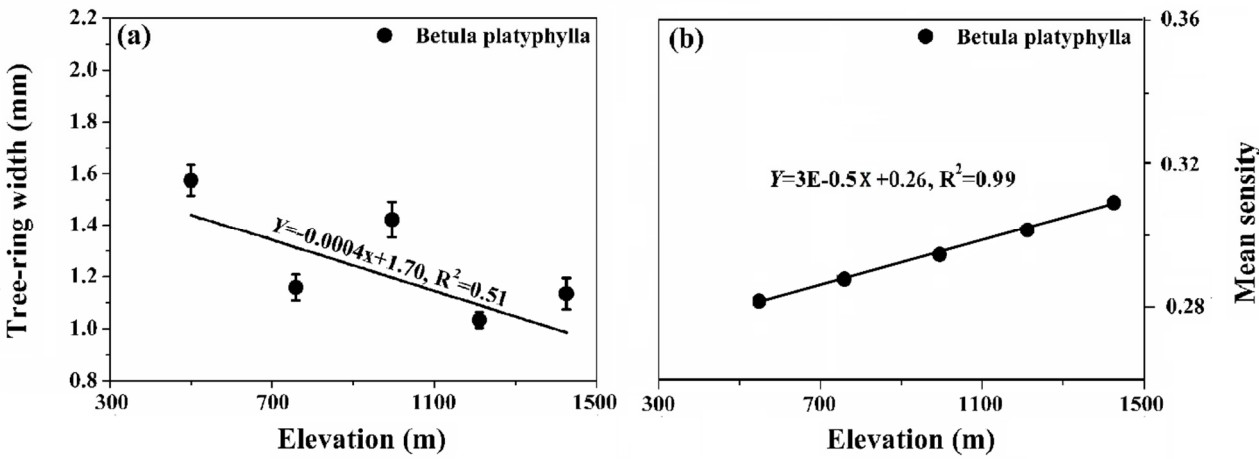

**Figure 5.** Variation in (**a**) mean tree-ring width (TRW), and (**b**) mean sensitivity (MS) of the STD chronologies along altitudinal gradients.

### 3.2. Climate Variation

Figure 3 showed that the annual mean temperatures has risen rapidly between 1970 and 2016 in the study area, the annual mean temperature rose by 0.22 °C every 10 years ($r = 0.71$, $p < 0.01$), and the annual mean temperature from 1970 to 2016 was 1.1 °C higher than that of 1946–1969 (Figure 3b). The annual mean maximum temperature has increased by 0.11 °C per 10 years ($r = 0.39$, $p < 0.05$), and the annual mean maximum temperature between 1970 and 2016 was 1.3 °C higher than that of 1946–1969 (Figure 3c). The rise in the annual mean minimum temperature was the most significant, with an increase of 0.37 °C every 10 years ($r = 0.81$, $p < 0.01$) (Figure 3a). Meanwhile, compared with the periods of 1946–1969, the average annual mean minimum temperature in 1970–2016 increased by 1.5 °C. The mean annual total precipitation in 1946–1969 and 1970–2016 was 824 mm and 821 mm, respectively, and the annual total precipitation during the period 1970–2016 displayed no significant variation. However, the PDSI value for the period 1970–2016 was 0.7 lower than the value from 1946 to 1969 (Figure 3d).

### 3.3. Climate–Growth Relationship

Correlation analysis showed that the relationships between the radial growth and climatic factors of *BP* trees were complicated and varied with altitude and time periods (Figure 6). During the period from 1946 to 2016, the radial growth of *BP* had a significant positive correlation with the monthly mean temperature (T) in May at all five locations, while the radial growth of *BP* at the locations of E550, E760, E995/E1210, and E1425 had a significant positive correlation with the monthly mean maximum temperature (Tmax)/the

monthly mean minimum temperature (Tmin) in May (Figure 6a). Meanwhile, significant positive correlations with T and Tmin were found for the previous year's (the previous year was abbreviated as t-1) September at E1210 and E1425, and significant positive correlations with T and Tmax occurred for October at E550 and E760 (Figure 6a). In addition, significant negative correlations with monthly total precipitation (Pre) occurred for March at E550, E760, and E995 (Figure 6a). *BP* tree growth has no significant correlation with PDSI (Figure 6a).

The climate/growth relationships between 1946 and 1969 were significantly different from that during 1946–2016 (Figure 6a,b). During 1946–1969, the growth of *BP* trees was not sensitive to temperature changes at sites E1210 and E1425; significant correlations were only found at site E1425 for Pre and PDSI in September (t-1) and October (t) (Figure 6b). The *BP* trees' radial growth was significantly impacted by monthly mean temperatures at site E550 and E760, and significant positive correlations were found during March–August (Figure 6b). Meanwhile, the significant positive correlations occurred for Tmax at site E550, for T, Tmin and Tmax at site E760 and for Tmin and T at site E995 in September (t-1) (Figure 6b). Significant positive correlations were also found for monthly mean temperatures in June at site E995 (Figure 6b). In addition, the radial growth at site E550 and E760 was significantly affected by Pre, with significant negative/positive correlations occurring for March(t)/November (t-1) (Figure 6b). Pre and PDSI in September (t-1) and October (t) had a significant positive effect on the tree growth of *BP* at site E1425 (Figure 6b).

**(a) 1946-2016**

| Month | Chronologies | E550 | | | | | E760 | | | | | E995 | | | | | E1210 | | | | | E1425 | | | | |
|---|---|---|---|---|---|---|---|---|---|---|---|---|---|---|---|---|---|---|---|---|---|---|---|---|---|---|
| | | Tmin | T | Tmax | Pre | PDSI | Tmin | T | Tmax | Pre | PDSI | Tmin | T | Tmax | Pre | PDSI | Tmin | T | Tmax | Pre | PDSI | Tmin | T | Tmax | Pre | PDSI |

Legend (correlation scale): −0.2 −0.1 −0.05 0 0.1 0.2 0.3 0.4

**Figure 6.** *Cont.*

**Figure 6.** Variability in the Pearson correlation coefficients between the STD/RES chronologies of *Betula platyphylla* and the monthly climatic variables during the phases of 1946–2016 (**a**), 1946–1969 (**b**), and 1970–2016 (**c**). Months 1–10: current year January-October; P9-P12: previous September-December. The open circles and solid circles represent negative and positive correlations, respectively. The circle represents the correlation-he larger the circle, the stronger the correlation. Gray shading indicates a significant correlation at the 0.05 level (2-tailed test). Unshaded circles indicate no significant correlation.

Compared with the periods of 1946–2016 and 1946–1969, the growth/climate relationships of 1970–2016 were completely different (Figure 6). During the periods of 1970–2016, the temperature had a more pronounced effect on the growth of *BP* trees of E1210 and E1425 (Figure 6). At site E1210, significant positive correlations were found with monthly mean temperatures in September (t-1), Tmin and T in May, Tmin in June, and Tmin and T during September (t) and October (t), while at site E1425, significant positive correlations occurred for T in September (t-1), monthly mean temperatures in May, Tmin in June, T and Tmax in September (t), and monthly mean temperatures in October (t) (Figure 6c). The growth of *BP* trees is not sensitive to climate change from September (t-1) to April at sites E550, E760, and E995 (Figure 6c). Meanwhile, compared with 1946–1969 and 1946–2016, the sensitivity of the radial growth of *BP* trees to temperature variations in May, September, and October was increased during the period 1970–2016 in these three sites (Figure 6). Significant positive correlations were found with T and Tmax in May for all the tree sites, Tmin and T in September (t) for E760, monthly mean temperatures in October(t) for E550, Tmax in October (t) for E760, and T and Tmax in October (t) for E995 (Figure 6c). In addition, contrary to 1946–1969, the temperatures in June to August during 1970–2016 displayed a negative effect on the radial growth. Significant correlations were observed for monthly mean temperatures in June and July at E550 and E760, while at E995, significant correlations were found for Tmin and T in June, monthly mean temperatures in July, and T and Tmax in August (Figure 6b,c). The Pre also had a significant impact on tree growth during 1970 and 2016, and significant positive correlations were observed in April at E1210 and E1425 and negative correlations were observed in May at E550 and E760 (Figure 6c).

During 1946 and 2016, the chronologies had significant positive correlations with T and Tmax for the early growing season (BG) at E550 and E760, and also had significant positive correlations with Tmax at E550 and monthly mean temperatures at E760 for the end of the growing season (EG) (Figure 7a). During 1946 and 1969, there were no significant climate/growth correlations observed at E995–E1425 (Figure 7b). At sites E550 and E760, *BP* growth had significant positive correlations with Tmax for the winter dormant period (WD) and with temperatures from the early growing season (BG) to the growing season (GS). Meanwhile, the ring-width indices at E550 had a significant positive correlation with Tmax in EG (Figure 7b). After 1970, the climate–growth relationships were obviously different, as the effects of BG and EG temperature on radial growth increased (Figure 7c). The radial growth was significantly positively correlated with temperatures in BG and EG at E550 and E760, with temperatures in EG at E1210 and E1425, with T and Tmax in BG and EG at E990, and also with Tmin in BG at E1210 and E1425 (Figure 7c). In addition, after 1970, the radial growth had a significant negative correlation with monthly mean temperatures in GS at E550–E995 (Figure 7c).

**Figure 7.** Variability in the Pearson correlation coefficients between the STD/RES chronologies of *Betula platyphylla* and the season period of the climatic variables during the phases 1946–2016 (**a**); 1946–1969 (**b**); and 1970–2016 (**c**). Season: winter dormant period (WD) (T < 0 °C), during previous year's November and current year's March; early growing season (BG) (Tmin < 5 °C), during April and May; growing season (GS), from June to August; and end of the growing season (EG) (Tmin < 5 °C), from the current year's September to current year. The open circles and solid circles represent negative and positive correlations, respectively. The circle represents the correlation-the larger the circle, the stronger the correlation. Gray shading indicates a significant correlation at the 0.05 level (2-tailed test). Unshaded circles indicate no significant correlation.

## 4. Discussion

### 4.1. Chronology Evaluation

Statistical analyses indicated a quite high quality for the STD chronologies (Table 1). The differences between altitudes were well represented in the chronology statistics. The values of the mean inter-series correlation and expressed population signal showed a certain amount of coherence between the series of the chronologies. The values of autocorrelation suggested that the influence of climatic factors in the previous year on the growth of *BP* trees is gradually strengthened with an increase in altitude. The average tree-ring width of white birch decreases linearly ($R^2$ = 0.51) with altitude (Figure 5a). Our findings are consistent with previous studies of other species (e.g., *Abies nephrolepis*, *Pinus koraiensis*, *Larix olgensis*, and *Picea koraiensis*) in Changbai Mountain [21,33]. The value of MS increased linearly with altitude ($R^2$ = 0.99) (Figure 5b), indicating that the variation in *BP* radial growth is greater at higher elevations. Our results showed that the MS (the value: 0.28–0.31) of *BP* was higher than that of other species studied in Changbai Mountain, such as *Betula ermanii* (0.23–0.34) [33,35], *Picea koraiensis* (0.13–0.17) [33], *Larix olgensis* (0.053–0.29) [32,33,37,49], *Pinus koraiensis* (0.11–0.20) [21,32,50,51], and *Fraxinus mandshurica* (0.11) [33].

### 4.2. The Relationship of Climate/Growth Along an Altitudinal Gradient from 1946 to 2016

Our results showed that, at low altitudes, *BP* has a higher correlation with temperature than precipitation (Figure 6a). At high altitudes (E1210–E1425), the growth of birch trees was significantly correlated with temperature (Figure 6a). The altitude-dependent growth/climate relationships were also reported in the *Betula ermanii* [35] and conifer species of CBM [21,32–34]. The current summer temperatures had no significant effect on the growth of *BP* trees; similar results were also found in Mount Norikura, central Japan [41], unlike the *BP* trees reported in Kamchatka [41]. At the low-elevation sites (E550–E995), the excessive March and May precipitation inhibited growth (Figure 6a). Adequate snowfall in March and May might meet the water requirements of *BP* in the early growing season, but it may delay the beginning of the growing season of the *BP* forest in Changbai Mountain. Many researchers have also found that high precipitation in March–May inhibits tree growth in mountain ecosystems [35,52]. It was found that the current May temperatures has a positive effect on the radial growth of *BP* at all altitudes (Figure 6a). Warmer spring temperatures may promote the emergence or germination of leaves earlier, resulting in better growth during prolonged growth seasons [3]. In contrast to our results, Alexander Gradel et al. (2017) [39] found that the growth of white birch (*Betula platyphylla* Suk.) was negatively correlated with the temperatures in May in Altansumber (49°29′07.29″ N; 105°31′30.36″ E), Northern Mongolia, and this relationship may be due to a water deficiency in spring [39]. Our results indicated that the temperature in September of the previous year was also a key factor affecting the radial growth of *BP* in high-altitude areas (E1210–E1425) (Figure 6a). This result was also observed in the study of the relationship between the growth of *Pinus koraiensis* Sieb. et Zucc. trees and climatic factors in Changbai Mountain [21]. The warmer conditions at the end of the growing season of the previous year helps the accumulation of more carbohydrates, which is beneficial for the growth of trees in the next year [3]. Our results indicated that the amount of carbohydrate storage in the previous year had a great impact on the growth of birch trees in the current year in the colder environments. In addition, *BP* stands at low altitudes were mainly responding positively to October temperatures in the current year (Figure 6a). In autumn, warmer conditions may prompt *BP* to continue photosynthesis [53]. Furthermore, the effects of the current spring and growing season temperatures on the radial growth of trees were complex and site-dependent (Figure 7a). At lower elevations, temperature during the early growing season and at the end of the growing season displayed positive correlations with tree growth, and there is no significant correlation at high altitudes (Figure 7a). Our results confirmed the following general conclusion: The relationship of climate/growth varies with altitude [52,53]. This agreed well with previous findings in this region [21,33].

### 4.3. Temporal Stability of the Climate–Growth Relationship

The results revealed that long-term sensitivity of *BP* growth to climate factors changed strongly along an altitudinal gradient, and the growth/climate relationship of *BP* trees showed temporal instability (Figure 7). This phenomenon has been observed in the study of the response of *Pinus koraiensis* Sieb. et Zucc. [21] and *Larix olgensis* Henry [37] growth to climate change at different altitudes in Changbai Mountain. Our results showed that, at high-altitude areas, before 1970 (Figures 6b and 7b), growth was not impacted by the current climate (Figure 6b). However, after 1970, the region has gradually warmed up (Figure 3), and the sensitivity of high-altitude *BP* trees to temperature has generally increased (Figures 6c and 7c). Tree growth displayed significant positive correlations with temperature in the early growing season (BG) and the end of the growing season (EG) (Figure 7c). Therefore, the warming after 1970 would promote the growth of *BP* trees in these areas.

At low-altitude areas, compared with the past, after 1970, the impact of precipitation on the growth of birch trees in March was gradually reduced, and the impact of precipitation in May was strengthened (Figure 6c). In addition, before 1970, the growth of trees was significantly positively correlated with the temperature from March to August (t) and from

September to November (t-1) (Figure 6b). However, after 1970, as the temperature rose, the number of months in which the growth of birch trees was positively correlated with temperature decreased with time (Figure 6c). Significant positive correlations between tree growth and temperature were found only in May (t), September (t), and October, while the correlation coefficients in the growing season have become negative (Figure 6c). The growth period of trees in CBM is from June to August. Therefore, as the temperature continues to rise, the growth of *BP* trees at low altitudes will be inhibited. The correlation results for monthly climate variables were also verified by results related to seasonal climate variables (Figure 7c).

As mentioned above, under continued warming in the future, the *BP* stands in Changbai Mountain will move to higher altitudes. Similar results were also obtained by Yu et al. (2013) and Wang et al. (2013) [21,34], who suggested that the radial growth of Korean pine increased at higher elevations, while decreased at low elevations under climate change characterized by warming and drought in Changbai Mountain. In addition, climate change can also cause changes in the distribution of tree species at certain latitudes in northeast China [22,27].

## 5. Conclusions

Temperature is a key factor affecting the growth of *BP* trees in Changbai Mountain. Although the growth of *BP* trees at all sampling points was significantly positively correlated with the temperature in May, in the period 1946–2016, the growth of birch trees distributed at high altitudes and low altitudes had different response patterns to climate change. Higher precipitation in March (t) and May (t) at low altitudes inhibits radial growth and warmer October (t) temperature accelerates radial growth, while the warmer September (t-1) temperature at high altitudes promotes radial growth. Furthermore, the results revealed that long-term sensitivity of *BP* growth to climate factors changed strongly along an altitudinal gradient, and the growth/climate relationships of *BP* trees showed temporal instability. Thus, the rise in temperature will inhibit the radial growth of *BP* at lower elevations and promote the growth of *BP* trees at higher elevations, dating from 1970. In the context of global climate change, the results will help simulate the radial growth and distribution dynamics of *BP* under various environmental conditions.

**Author Contributions:** J.Z., and S.H. conceived the idea; Y.J. and Y.C. performed the experiment; Y.J. and Y.C. analyzed data; Y.J. wrote the manuscript; L.H. writing—review and editing; J.Z. and S.H. reviewed and approved the final manuscript. All authors have read and agreed to the published version of the manuscript.

**Funding:** The research was funded by the National Key Research and Development Program of China (2016YFA0600800), Academy of Changbai Mountain Science (Grant No. 2017-03) and the Educational Department of Liaoning Province (Nos. LQN 201905).

**Data Availability Statement:** The data presented in this study are available on request from the first author.

**Conflicts of Interest:** The authors declare no conflict of interest.

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
