# Peer review of "Spatial Variation and Temporal Instability in the Growth/Climate Relationship of White Birch (Betula platyphylla Suk) in the Changbai Mountain, China"

_forests, doi:10.3390/f12050589_

Round 1

Reviewer 1 Report

Authors aimed to examine the spatial variation and temporal stability of the growth/climate relationship of Betula platyphylla. Authors used the dendrochronological method to analyze the radial-growth/climate relationship of the B. platyphylla trees along the altitudinal gradient in Changbai Mountain of northeast China.

Overall comment: In my opinion, the presented theme meets the aims of Forests MDPI Journal. The results are quite interesting. However I have some other doubts. The concern the beginning of the study (abstract and introduction) and are assosciated with the novelty of the study. In my opinion, the results presented in the abstract and introduction suggest that the work is already in line with the generally known trends in climate changes. The region may be new, although it has been shown to be a fairly heavily studied area. Also the species may be new (B. platyphylla) in the studied region. Authors state that this is the first study about BP in the studied region "So far, the research on the response of the growth of BP trees to climatic factors in the northeastern China has not been carried out". But, my question is, are there any other scientific research about BP in other regions of Asia (Japan, Korea, Siberia) concerning similar studies? It would be more helpful to point clearly the importance of the work and would emphasize the novelty of the study besides the new area of CBM. The lack of such information raises questions about the novelty, and makes the research a local study that only extends previous knowledge. However, such doubts can be easily explained in the revised version. 

In the abstract, in order to show the importance of the topic and the scientific understanding, I recommend to indicate the scope of the years studied.

Moreover, I recommend to revise the whole manuscript to delete many  technique and some minor language flaws.

Below I listed detalied comments:

Lines:

16 - please unify the the phrase Betula platyphylla by using abbrev. BP which was already used in l.15, 169, 

18-22 - I advise to add some detailed data to the sentences presenting major results of the study (i.e. about the altitude, temp, precipit.)

26- rewrite the word "radial" by using capital letter (Radial growth).

32-33 - in the sentence "in species phenology, biomass, diversity, distribution and ecosystem processes..." add a comma before the word "and". See also line 62.

59-60 - the sentence: "It has economic value, can reach up to 27 meters, and can survive for 120-140 years with the habitat types of light  taiga" should be changed i.e. to "It has an economic value, can reach up to 27 meters height, and can survive for 120-140 years within the light taiga habitat".

95 - here should be "in November 2016". It was to wordy when used "of".

104 - Why the sampling sites were established all on the north slope?

111-112 - I advise to rewrite this part of the sentence: "the selected White birch forests were natural and open forests, there were no signs of recent fires or human disturbances" on to: the selected White birch forests were  natural and open forests with no signs of recent fires or human disturbances.

142- Is the title of Table 1 correct? I found that there are presented the characteristics of Larix gmelinii...

Moreover in the fifth line of the Table 1 "No. of cores", the values should be formatted strictly as the other text in the table (center allignment).

149 - check the dot at he the end of the sentence, as well as the font size  of link to the website.

167 - correct comma position

175 - correct the phrase "tocurrent"

In 3.2. subchapter, I recommend to use abbreviations for annual mean temperature, annual mean maximum temperature, annual mean minimum temperature, mean annual total precipitation, annual total precipitation.

Moreover in line 213 add a space between values of MATP and mm units.

227 - technique flaw - look a the end of the sentence

268 - correct the phrase "monthlymean"

290 - add a space between bracket and the "chronologies"

4.1. subchapter. I would see here also some additional information about scientific research about BP in other regions of Asia (Japan, Korea, Siberia)? Are there any similar studies about the species in question?  It would be more helpful to discuss the results. At this point, the discussion concerns the obtained results without reference to the results of work in other regions of the country or Asia.

424 - add a dot at the end of line.

Author Response

Reviewer 1

Overall comment: In my opinion, the presented theme meets the aims of Forests MDPI Journal. The results are quite interesting. However, I have some other doubts. The concern the beginning of the study (abstract and introduction) and are assosciated with the novelty of the study. In my opinion, the results presented in the abstract and introduction suggest that the work is already in line with the generally known trends in climate changes. The region may be new, although it has been shown to be a fairly heavily studied area. Also the species may be new (B. platyphylla) in the studied region. Authors state that this is the first study about BP in the studied region "So far, the research on the response of the growth of BP trees to climatic factors in the northeastern China has not been carried out".

  1. But, my question is, are there any other scientific research about BP in other regions of Asia (Japan, Korea, Siberia) concerning similar studies? It would be more helpful to point clearly the importance of the work and would emphasize the novelty of the study besides the new area of CBM. The lack of such information raises questions about the novelty, and makes the research a local study that only extends previous knowledge. However, such doubts can be easily explained in the revised version. 

Response: Thank you very much for pointing it out. The other scientific research about BP was added in the line 86-88, 340-341, and 351-355.

  1. In the abstract, in order to show the importance of the topic and the scientific understanding, I recommend to indicate the scope of the years studied.

Response: Thank you very much for pointing it out. The scope of the years was added in line 33

  1. Moreover, Irecommend to revise the whole manuscript to delete many technique and some minor language flaws.

Response: Thank you very much for pointing it out. The article has been revised in accordance with the reviewer’s comments

Below I listed detalied comments:

Lines:

  1. 16 - please unify the the phrase Betula platyphylla by using abbrev. BP which was already used in l.15, 169, 

Response: Thank you very much for pointing it out. This was revised in line 35, 328, 357

  1. 18-22 - I advise to add some detailed data to the sentences presenting major results of the study (i.e. about the altitude, temp, precipit.)

Response: Thank you very much for pointing it out. This was revised in line 39-41.

  1. 26- rewrite the word "radial" by using capital letter (Radial growth).

Response: Thank you very much for pointing it out. This was revised in line 48.

  1. 32-33 - in the sentence "in species phenology, biomass, diversity, distribution and ecosystem processes..." add a comma before the word "and". See also line 62.

Response: Thank you very much for pointing it out. This was revised in line 54 and 89.

  1. 59-60 - the sentence: "It has economic value, can reach up to 27 meters, and can survive for 120-140 years with the habitat types of light  taiga" should be changed i.e. to "It has an economic value, can reach up to 27 meters height, and can survive for 120-140 years within the light taiga habitat".

Response: Thank you very much for pointing it out. This was revised in line 84-86.

  1. 95 - here should be "in November 2016". It was to wordy when used "of".

Response: Thank you very much for pointing it out. This was revised in line 128.

  1. 104 - Why the sampling sites were established all on the north slope?

Response: Thank you very much for pointing it out. In order to eliminate the influence of slope differences on the growth of trees, the trees were selected with the same slope.

Reference:

Kujansuu J, Yasue K, Koike T, et al. Responses of ring widths and maximum densities of Larix gmelinii to climate on contrasting north- and south-facing slopes in central Siberia[J]. Ecological Research, 2007, 22(4):582-592.

  1. 111-112 - I advise to rewrite this part of the sentence: "the selected White birch forests were natural and open forests, there were no signs of recent fires or human disturbances" on to: the selected White birch forests were  natural and open forests with no signs of recent fires or human disturbances.

Response: Thank you very much for pointing it out. This was revised in line 140-142.

  1. 142- Is the title of Table 1 correct? I found that there are presented the characteristics of Larix gmelinii...Moreover in the fifth line of the Table 1 "No. of cores", the values should be formatted strictly as the other text in the table (center allignment).

Response: Thank you very much for pointing it out. This was revised in line 616 and the fifth line of the Table 1

  1. 149 - check the dot at he the end of the sentence, as well as the font size of link to the website.

Response: Thank you very much for pointing it out. This was revised in line 173.

  1. 167 - correct comma position

Response: Thank you very much for pointing it out. This was revised in line 195.

  1. 175 - correct the phrase "to current"

Response: Thank you very much for pointing it out. This was revised in line 206.

  1. In 3.2. subchapter, I recommend to use abbreviations for annual mean temperature, annual mean maximum temperature, annual mean minimum temperature, mean annual total precipitation, annual total precipitation. Moreover in line 213 add a space between values of MATP and mm units.

Response: Thank you very much for pointing it out. Because only this part mentions these parameters. So, we think the style of non-abbreviations may be appropriate, and the space was added in line 234.

  1. 227 - technique flaw - look at the end of the sentence

Response: Thank you very much for pointing it out. This was revised in line 250.

  1. 268 - correct the phrase "monthlymean"

Response: Thank you very much for pointing it out. This was revised in line 289.

  1. 290 - add a space between bracket and the "chronologies"

Response: Thank you very much for pointing it out. This was revised in line 657-658.

  1. 4.1. subchapter. I would see here also some additional information about scientific research about BP in other regions of Asia (Japan, Korea, Siberia)? Are there any similar studies about the species in question?  It would be more helpful to discuss the results. At this point, the discussion concerns the obtained results without reference to the results of work in other regions of the country or Asia.

Response: Thank you very much for pointing it out. Similar studies were cited and integrated in the line 86-88, 340-341, and 351-355.

  1. 424 - add a dot at the end of line.

Response: Thank you very much for pointing it out. This was revised in line 478.

Reviewer 2 Report

The paper presents study of a local relevance which itself is not a wrong thing, but despite this locality it should be placed in broader context, namely as a case study - Authors forget about it omitting many references and giving no literature background (no references out of China - so why publish it internationally??) to the study that itself is not novel - changes of the growth/climate relationship along the altitude was a subject of many research world-wide, many of them easily found with a simple google-search

the manuscript is prepared with many flaws in editing:

L34 what exactly has the rapidest warming?

L52 deciduous and broad-leaved are the same

L56-57 only Latin name in italics

L83-L84: northeast China & Northeast China,

L119 - November 2016 repeated uneccessarily here

L182 - dont describe what is in the table - the caption does it

L204 - Figure 5 does not show what you write here

L314 wrong reference citation Yu et al. 2011

l324 figure 7a does not refer to other species

and consistency:

study sites must be labeled the same way all over the paper: BP1-BP5 in figure 1, E550m -E1425m in section 3.3; E550 -E1425 - figures 6-7

"previous" issue - use the word previous all the time, t-1 does not correspond with P in figures 6 and 7

major problems for me

L92 - this Ta and P refer to what? time or sites

L133 why (what for) did you do RES and ARS chronologies???

table 1 - these are not stats for residual chronologies - in RES chrono autocorrelation is removed, here you have high values

L172-175 why (what for) do you introduce that classification here, no reason presented here

L186 you dont know what this SD refers to, do you? it's standard deviation (measure of variability) of tree-ring widths

L193-194 - this influence of the previous year is wrong, you should get rid of it, that is why one develops RES chronos

L299-300 c'mon, you did not analysed the effectiveness of the tree-ring analyses

L356 you have to decide whether growth of BP was sensitive or not to climate change; you mix change with normal fluctuations I guess

THE MOST IMPORTANT

if you detected trend in climate data, you should detrend it as well, and than correlate it

other issues

L104-105 use here BP1 and other labels instead of altitudes

L125 accuracy not resolution

L135 what is EPS, it should be explained and reference given

Author Response

Reviewer 2

The paper presents study of a local relevance which itself is not a wrong thing, but despite this locality it should be placed in broader context, namely as a case study - Authors forget about it omitting many references and giving no literature background (no references out of China - so why publish it internationally??) to the study that itself is not novel - changes of the growth/climate relationship along the altitude was a subject of many research world-wide, many of them easily found with a simple google-search

Response

the manuscript is prepared with many flaws in editing:

  1. L34 what exactly has the rapidest warming?

Response: Thank you very much for pointing it out. This was revised in line 55-56.

  1. L52 deciduous and broad-leaved are the same

Response: Thank you very much for pointing it out. The word broad-leaved was deleted in line 76.

  1. L56-57 only Latin name in italics

Response: Thank you very much for pointing it out. This was revised in line 80.

  1. L83-L84: northeast China & Northeast China,

Response: Thank you very much for pointing it out. This was revised in line 115.

  1. L119 - November 2016 repeated uneccessarily here

Response: Thank you very much for pointing it out. November 2016 was revised in line 149.

  1. L182 - dont describe what is in the table - the caption does it

Response: Thank you very much for pointing it out. This sentence was deleted.

  1. L204 - Figure 5 does not show what you write here

Response: Thank you very much for pointing it out. Figure was revised by Figure 3 in line 223.

  1. L314 wrong reference citation Yu et al. 2011

Response: Thank you very much for pointing it out. This was revised in line 330.

  1. l324 figure 7a does not refer to other species and consistency:

study sites must be labeled the same way all over the paper: BP1-BP5 in figure 1, E550m -E1425m in section 3.3; E550 -E1425 - figures 6-7

"previous" issue - use the word previous all the time, t-1 does not correspond with P in figures 6 and 7

Response: Thank you very much for pointing it out. Similar studies were cited and integrated in line 341-342. In addition, the study sites were labeled the same way all over the paper.

major problems for me

  1. L92 - this Ta and P refer to what? time or sites

Response: Thank you very much for pointing it out. In line 124-126, the Ta is the annual average temperature and the P is the annual precipitation, while T in this paper is monthly mean temperature, Pre is monthly total precipitation.

  1. L133 why (what for) did you do RES and ARS chronologies??? Response: Thank you very much for pointing it out. To eliminate the effects of stand dynamics, age, and any other non-climate-related growth variation, the cross-dated tree-ring data were detrended using three different techniques—negative exponential (EXP), regional curve standardization (RCS), via ARSTAN [47]. After these, three kinds of chronologies were generated from ARSTAN: STD-standard, RES-residual and ARS-autoregressive. The STD chronology with an EXP detrending was the best chronology because it contained more low frequency signals.
  2. table 1 - these are not stats for residual chronologies - in RES chrono autocorrelation is removed, here you have high values

Response: Thank you very much for pointing it out. In this paper, “White birch STD” chronologies were selected rather than “Larix gmelini RES”. This is the negligence of the writer.

  1. L172-175 why (what for) do you introduce that classification here, no reason presented here

Response: Thank you very much for pointing it out. The reason was added in line 199-200.

  1. L186 you dont know what this SD refers to, do you? it's standard deviation (measure of variability) of tree-ring widths

Response: Thank you very much for pointing it out. This was revised in line 216.

  1. L193-194 - this influence of the previous year is wrong, you should get rid of it, that is why one develops RES chronos

Response: Thank you very much for pointing it out. This study uses STD chronology instead of RES chronology. The RES appeared in the caption of Table 1 was because of a writing error.

  1. L299-300 c'mon, you did not analysed the effectiveness of the tree-ring analyses

Response: Thank you very much for pointing it out. This sentence was deleted.

  1. L356 you have to decide whether growth of BP was sensitive or not to climate change; you mix change with normal fluctuations I guess

Response: Thank you very much for pointing it out. This was revised in line 384-386.

THE MOST IMPORTANT

  1. if you detected trend in climate data, you should detrend it as well, and than correlate it

Response: Thank you very much for pointing it out. This was revised in line 183-185 and Figure 6 and Figure 7.

other issues

  1. L104-105 use here BP1 and other labels instead of altitudes

Response: Thank you very much for pointing it out. This was revised in line 133-134.

  1. L125 accuracy not resolution

Response: Thank you very much for pointing it out. This was revised in line 157.

  1. L135 what is EPS, it should be explained and reference given

Response: Thank you very much for pointing it out. This was revised in line 167.

Reviewer 3 Report

In the manuscript “Spatial variation and temporal instability in the growth/climate relationship of White birch (Betula platyphylla) in the Changbai Mountain, China”, Jiang and colleagues analyze five chronologies of Betula platyphylla Sukaczev sampled along an altitudinal transect of about 925 m in Changbai Mountain, North-East China. Basic dendrochronological analyses were performed on a well replicated datasets that span between 66 and 90 years. The results highlight a change of the tree’s climate sensitivity along the transect in the last fifty years, with specimens located at highest altitudes recording an enhanced correlation with temperature in the recent period. Conversely, the low altitude specimens record a lowering in correlation values. The authors conclude that, under the effect of the ongoing climate change, forests of Betula platyphylla Sukaczev of northeastern China will undergo through changes in the distribution range in altitude and latitude.

In my opinion, the manuscript is in line with the aims of the journal Forests reporting information about the impact of climate change on natural secondary forests. I personally find the manuscript almost clear but with minor flaws in the organization of the text; the scientific design is well depicted with a satisfying number of specimens per plot and a quite homogeneous distribution of the plots along the altitudinal transect. Obtained results are encouraging albeit, in my opinion, should be more deeply investigated. Thus, in my opinion, the manuscript is interesting and worthy to be considered for publication but only after an integration of what I consider, in this version, a possible methodological issue.

Finally, I think that the text could benefit from both a careful reading by the authors and an English grammar and spells checking.

Please find my concerns in the attached file (Please note that line numbers refer to the *.pdf version of the manuscript).

Author Response

Reviewer 3:

In the manuscript “Spatial variation and temporal instability in the growth/climate relationship of White birch (Betula platyphylla) in the Changbai Mountain, China”, Jiang and colleagues analyze five chronologies of Betula platyphylla Sukaczev sampled along an altitudinal transect of about 925 m in Changbai Mountain, North-East China. Basic dendrochronological analyses were performed on a well replicated datasets that span between 66

and 90 years. The results highlight a change of the tree’s climate sensitivity along the transect in the last fifty years, with specimens located at highest altitudes recording an enhanced correlation with temperature in the recent period. Conversely, the low altitude specimens record a lowering in correlation values. The authors

conclude that, under the effect of the ongoing climate change, forests of Betula platyphylla Sukaczev of northeastern China will undergo through changes in the distribution range in altitude and latitude. In my opinion, the manuscript is in line with the aims of the journal Forests reporting information about the impact

of climate change on natural secondary forests. I personally find the manuscript almost clear but with minor flaws in the organization of the text; the scientific design is well depicted with a satisfying number of specimens per plot and a quite homogeneous distribution of the plots along the altitudinal transect. Obtained results are encouraging albeit, in my opinion, should be more deeply investigated. Thus, in my opinion, the manuscript is interesting and worthy to be considered for publication but only after an integration of what I consider, in this

version, a possible methodological issue.

Finally, I think that the text could benefit from both a careful reading by the authors and an English grammar and

spells checking.

Please find below my concerns (Please note that line numbers refer to the *.pdf version of the manuscript).

MAJOR  COMMENTS

The Authors in their manuscript analyze the chronologies splitting them in two uneven parts, calculating Pearson’s coefficients between predictor and predictand and valuating them as an index of sensitivity of the species to climate. It is correct but, the fact that an increasing trend is visible only in the two chronologies representative of the high-altitude plots and the fact that, at the same time, they represent the oldest chronologies, lead to think that the increasing trend observable at the end of the chronologies could be a bias induced by the chosen standardization method (Negative Exponential Curves). Thus, I suggest to test if other standardization methods on the same datasets return similar results. Moreover, also the shift from positive to negative correlation values in the low altitude stands (even if the entity of this shift is never reported in the manuscript) seems to be induced by the different long-term trend showed by the TRW data and temperature. Thus, for a more comprehensive study, I suggest to consider also the results of other standardization methods applied at the same datasets and, moreover, would be interesting to know if the observed sensitivity, and related changes, is linked only to the mid-frequencies or if it is related also to the high-frequencies. Regarding the latter, it would be interesting to know if the analyzed species is sensitive to high-frequencies climate variations in the past and if the sensitivity is changed under the pressure of the ongoing climate change at the different altitudes as well as the sensitivity to mid-frequency already analyzed. With this aim, a more attention has to be given to the RES chronologies.

Response: Thank you very much for pointing it out. To eliminate the effects of stand dynamics, age, and any other non-climate-related growth variation, the cross-dated tree-ring data were detrended using two different techniques—negative exponential (EXP), regional curve standardization (RCS), via ARSTAN. After these, three kinds of chronologies were generated from ARSTAN: STD-standard, RES-residual and ARS-autoregressive. The STD chronology with an EXP detrending was the best chronology because it contained more low frequency signals (Figure 4)

MINOR COMMENTS

  1. Lines 40–43: It is not clear to me how the change in the radial growth could be the cause of a change in the distribution of the species. Please consider rephrasing the sentence.

Response: Thank you very much for pointing it out. The sentence was revised in line 62-65.

  1. Lines 92–93: Please consider to rephrase this sentence considering that the mean annual temperature and precipitation should have specific values and, eventually, the associated errors. Moreover, mean values should be calculated on a period of time of at least 30 years. Consider to report also this last information for completeness of the description. Contrarily, if the mean temperature values are referred to the lowest and highest altitude, it should be reported as well for clarity.

Response: Thank you very much for pointing it out. This was revised in line 124-126.

  1. Line 97: Consider removing the comma as thousand separators for consistency with the following values.

Response: Thank you very much for pointing it out. This was revised in line 130.

  1. Lines 128–131: The Authors state “For the old trees selected were little affected by competition, negative exponential curve (EXP) was used to standardize the raw tree-ring data [45] to eliminate the effects of stand dynamics, age, and any other non-climate-related growth variation.”. Nevertheless, no information about what is considered “old” and what is considered “young” is reported neither the type of standardization applied to the “young” specimens. Moreover, the acronym for the Negative Exponential Curve is not used anymore along the text thus please consider to remove it.

Response: Thank you very much for pointing it out. The sentence was revised in line 159-163.

  1. Line 135: Please check the figure reference and order, I think that Authors would refer to Figure 4.

Response: Thank you very much for pointing it out. Figure 3 was revised by Figure 4 in line 166.

  1. Line 136: It is not clear to me how the EPS could confirm the maximum length of reliable chronology. Please consider to rephrase.

Response: Thank you very much for pointing it out. This was revised in line 167.

  1. Lines 136–137: Please consider to remove the sentence “The chronology statistics was shown in Table1”, it is inappropriate for Materials and Methods section and already, and properly, states in the results section.

Response: Thank you very much for pointing it out. This sentence was deleted.

  1. Lines 147–149: Please add the proper citation to the dataset. Harris at al. 2014.

Response: Thank you very much for pointing it out. The reference was added in line 173.

  1. Line 154: Songjiang and Fusong meteorological stations are not reported in Figure 1. Please consider to add their location at that Figure.

Response: Thank you very much for pointing it out. The meteorological stations were added in the Figure 1.

  1. Lines 155–157: Tautological phrases, please remove one of them.

Response: Thank you very much for pointing it out. This was revised in line 180-181.

  1. Lines 160–162: Please consider to move this sentence to the more appropriate Results section.

Response: Thank you very much for pointing it out. We think it might be appropriate to place this sentence in this part. Because we want to explain why we want to do the relevant analysis after 1970.

  1. Line 175: Please correct the typo “tocurrent” in “to current”.

Response: Thank you very much for pointing it out. This was revised in line 206.

  1. Line 184: Please consider to remove the unnecessary equation.

Response: Thank you very much for pointing it out. The equation was deleted in line 214.

  1. Lines 185–186: The RBAR statistic does not indicate the reliability of a chronology. Please correct this sentence.

Response: Thank you very much for pointing it out. This sentence was revised in line 214-215.

  1. Lines 188–191: Please consider to rephrase or delete the sentence since has been proved that the MS parameter “is a confusing and ambiguous statistic for describing the variations in tree growth” and is largely dependent from the autocorrelation and standard deviation of a series (Bunn et al., 2013). Moreover, the authors in their statement refer to [47] but the authors of that paper refer to another work (in Russian) for that sentence. Thus, please refer to the original work if the content is known or delete it.

Response: Thank you very much for pointing it out. This sentence was deleted.

  1. Lines 192–193: As same as comment to line 184.

Response: Thank you very much for pointing it out. This sentence was deleted.

  1. Line 204: I think that the Authors want to refer to Figure 3 and not 5. Please correct.

Response: Thank you very much for pointing it out. This was revised in line 223.

  1. Line 215: “-0.7 lower” is pleonastic, please consider to remove the symbol or the adverb.

Response: Thank you very much for pointing it out. This was revised in line 236-237.

  1. Line 216: Perhaps the Authors mean “between 1927 and 1969”. Please check the English spell (along the manuscript) and the period in this sentence.

Response: Thank you very much for pointing it out. This was revised in line 237, 255.

  1. Line 221 Henceforth: the name of the sampled sites are changed. Please be consistent.

Response: Thank you very much for pointing it out. This problem has been solved in this paper.

  1. Lines 220–224: Please rephrase this long and confusing sentence.

Response: Thank you very much for pointing it out. This sentence was revised in line 241-246.

  1. Lines 249–250: This concept was already reported at lines 240/241. Please remove it.

Response: Thank you very much for pointing it out. This sentence was deleted.

  1. Line 257: I think that the Authors mean September (t-1). Please correct it.

Response: Thank you very much for pointing it out. The September (t) was revised by September (t-1) in line 277.

  1. Line 258: I think that the Authors mean T and Tmax in September (t). Please correct it and check the correctness of other reported information along all the Results section.

Response: Thank you very much for pointing it out. This was revised in line 278.

  1. Line 268: “monthly mean” with space.

Response: Thank you very much for pointing it out. This was revised in line 290.

  1. Line 282: The parameter Tm was not declared before.

Response: Thank you very much for pointing it out. The “Tm” was revised by Tmax in line 306.

  1. Lines 309–310: Typos in species names, please correct them.

Response: Thank you very much for pointing it out. This was revised in line 325.

  1. Lines 315: “Fraxinus” with n.

Response: Thank you very much for pointing it out. This was revised in line 332.

  1. Lines 355–358: Please consider to rephrase this sentence. The growth is not sensible to climate change but to climate and the effect of climate change could be visible through changes in the growth rate or changes in sensitivity of the tree species.

Response: Thank you very much for pointing it out. This sentence was revised in line 384-386.

  1. Line 386: Perhaps the authors want to refer to 1927–2016 period.

Response: Thank you very much for pointing it out. This was revised in line 421.

  1. Figure 1: Please consider to add another longitude coordinate mark to make possible the calculation of the geographical longitude at each point of the map. Moreover, the latitude marks on the left are covered by the insets, please uncover them. Finally, if available, consider to add a DTM or an hillshade to the main map to give an idea of the elevation of the area.

Caption: consider to change “points” with “sites”.

Response: Thank you very much for pointing it out. The figure 1 was revised, and “points” was revised by “sites” in line 632.

  1. Figure 2: Please check the legend to be consistent with plot symbols. Caption: I suppose that the plot refers to the CRU data, thus, please consider to specify it and to add the proper citation (Harris et al. 2014).

Response: Thank you very much for pointing it out. The legend of Figure 2 was revised and the citation was added.

  1. Figure 3: Please consider to plot also the precipitation series since more attention was given to this parameter rather than PDSI.

Response: Thank you very much for pointing it out. The precipitation series was added in line Figure 3(e)

  1. Figure 4: I personally think that the number of individuals is more interesting and informative rather than the number of cores used in the chronologies. Please consider to add or switch to this parameter.

Caption: please consider to specify that the plots are referring to standard chronologies.

Response: Thank you very much for pointing it out. This was revised in Figure 4 and Table 1.

  1. Figure 6: Please provide the legend of the symbols, otherwise it is impossible to assign a value of correlation to each circle; the simple dimension could mean 1 as well as 0.2 without a legend. Moreover, I think that the header of the table is misleading since only one chronology starts in 1927, all the others are younger with the youngest that covers the period only between 1951 and 1969 (19 years) or 2016

Response: Thank you very much for pointing it out. The legend of the symbols was added. In addition, this paper does have the problem you pointed out. However, considering the short time span between 1951-1969 and the small change in the value of climate factors between 1927-1969, we used the data from 1927-1969.

  1. Figure 7: see comment to Figure 6.

Response: Thank you very much for pointing it out. The legend of the symbols was added.

  1. Table 1: Together with the number of cores for each site, could be useful to report also the number of trees.

Response: Thank you very much for pointing it out. This was revised in Table 1.

  1. Caption: at lines 135 and 182–183 is declared that STD was used for the analysis, but in Table 1 the information of the RES chronology of another tree species is reported. Please check the correctness of these information removing the redundant part of the caption.

Response: Thank you very much for pointing it out. This was revised in line 617

Round 2

Reviewer 1 Report

The revised version of the manuscript seems to be satisfactory. The authors responded to most of the comments. They supplemented the literature.
However, the work requires significant corrections of a proofreading nature. The authors did not avoid to the numerous flaws in the text. However, taking into account the superiority of the article's merit over editing flaws, I believe that the work can be accepted for publication. Below I present some of the flaws that I noticed at revised version. The entire manuscript should be read carefully to unify the phrases spelling and remove editing flaws.

Comments:

31- delete additional space between growth/climate

39- I would rather replace phrases i.e. (E550-E995) with 550-995 m a. s. l.,

41- as above,

104, 108,110- the BP abbrev. is given once in italics and the second time without ilalics. Please unify BP abbrev. in whole manuscript.

126- remove additional space in the first bracket, as well as space after the value of 88.4 (88.4%).

129-131- check the spelling the phrases of a.s.l in the mentioned lines,

144- give the meters abbrev. of trees height,

210- add a space after 550,

234- Unify by adding a dash or pause between date's ranges. Check the whole manuscript. Check also line 170.

240-244 - see the comment for lines 104, 108...,

244, 280- Use commas to separate words and word groups in a simple series of three or more items. Please revised the text.

352-356- I do not fully understand the sentence. Please rewrite it.

617-Table 1: chceck the alignment of the text.

641-644- correct the spelling of alttude, the abbrev. of meters should be separated by a space from values...

Author Response

Reviewer 1

The revised version of the manuscript seems to be satisfactory. The authors responded to most of the comments. They supplemented the literature. However, the work requires significant corrections of a proofreading nature. The authors did not avoid to the numerous flaws in the text. However, taking into account the superiority of the article's merit over editing flaws, I believe that the work can be accepted for publication. Below I present some of the flaws that I noticed at revised version. The entire manuscript should be read carefully to unify the phrases spelling and remove editing flaws.

Comments:

31- delete additional space between growth/climate

Response: Thank you very much for pointing it out. The additional space was deleted in Page 2 Line 31.

39- I would rather replace phrases i.e. (E550-E995) with 550-995 m a. s. l.,

Response: Thank you very much for pointing it out. “E550-E995” has revised as 550-995 m a. s. l., in Page 2 Line 39

41- as above,

104, 108,110- the BP abbrev. is given once in italics and the second time without ilalics. Please unify BP abbrev. in whole manuscript.

Response: Thank you very much for pointing it out. The BP abbrev. Were unified in whole manuscript.

126- remove additional space in the first bracket, as well as space after the value of 88.4 (88.4%).

Response: Thank you very much for pointing it out. The question has resolved in Page5 Line126.

129-131- check the spelling the phrases of a.s.l in the mentioned lines,

Response: Thank you very much for pointing it out. The “a.s.l” has revised as a. s. l in Page 5 Line 131.

144- give the meters abbrev. of trees height,

Response: Thank you very much for pointing it out. The word meters was revised by m in Page 6 line144

210- add a space after 550,

Response: Thank you very much for pointing it out. A space has added in Page 8 Line 211.

234- Unify by adding a dash or pause between date's ranges. Check the whole manuscript. Check also line 170.

Response: Thank you very much for pointing it out. This was unified unified in whole manuscript.

240-244 - see the comment for lines 104, 108...,

Response: Thank you very much for pointing it out. The question was resolved in Page 9 line 240-244.

244, 280- Use commas to separate words and word groups in a simple series of three or more items. Please revised the text.

Response: Thank you very much for pointing it out. This was revised in Page 9 line 244 and Page 11 Line 280.

352-356- I do not fully understand the sentence. Please rewrite it.

Response: Thank you very much for pointing it out. The sentence has revised as “In contrast to our results, Alexander Gradel et al., 2017 [40] found that the growth of White birch (Betula platyphylla Suk) was negatively correlated with the temperatures in May in Altansumber (49°29′07.29″N; 105°31′30.36″E) of Northern Mongolia, this relationship may be due to a water deficiency in spring” in Page 13 Line 352-356.

617-Table 1: chceck the alignment of the text.

Response: Thank you very much for pointing it out. This was revised in Page 21 Line 616-617.

641-644- correct the spelling of alttude, the abbrev. of meters should be separated by a space from values...

Response: Thank you very much for pointing it out. This was revised in Page 22 Line 640-643.

Reviewer 2 Report

Authors in general improved the paper - there are still some flaws, but mainly of editorial character

Author Response

Dear editor, thank your very much for your help during the revision process.

The paper was revised based on the suggestions of  the reviewers

Reviewer 3 Report

In the amended version of the manuscript “Spatial variation and temporal instability in the growth/climate relationship of White birch (Betula platyphylla) in the Changbai Mountain, China”, Jiang and colleagues did not provide any significant change compared to the previous version. The additional analyses required by me and the other reviewer were not performed or not reported, as exception of few words in the materials and methods section, not followed by obtained results and discussion of them. The Authors only added a STD RCS chronology to the Figure 4 and they wrote that the EXP chronology is the best while all the rest of the text remains basically unvaried. Moreover, they overlook on the fact that RCS can’t be applied to a dataset such that here presented without adding important bias in mid- and low-frequency; see major comment for more details and references. Thus, in my opinion, the manuscript is still interesting and worthy to be considered for publication but only after a deep and critical revision and integration of what I still consider methodological issues (i.e. the lack of high- and mid-frequency analysis separated to understand which component of the climate change most affects the tree growth).

Finally, I think that the text could benefit from both a careful reading by the Authors and an English grammar and spells checking by mother tongue reader or professional English editorial service.

Please find below my concerns (Please note that line numbers refer to the *.pdf version of the manuscript).

Author Response

Reviewer 3

In the amended version of the manuscript “Spatial variation and temporal instability in the growth/climate relationship of White birch (Betula platyphylla) in the Changbai Mountain, China”, Jiang and colleagues did not provide any significant change compared to the previous version. The additional analyses required by me and the other reviewer were not performed or not reported, as exception of few words in the material and method section, not followed by obtained results and discussion of them. The Authors only added a STD RCS chronology

to the Figure 4 and they wrote that the EXP chronology is the best while all the rest of the text remains basically unvaried. Moreover, they overlook on the fact that RCS can’t be applied to a dataset such that here presented without adding important bias in mid- and low-frequency; see major comment for more details and references. Thus, in my opinion, the manuscript is still interesting and worthy to be considered for publication but only after a deep and critical revision and integration of what I still consider methodological issues (i.e. the lack of high- and mid-frequency analysis separated to understand which component of the climate change most affects the tree growth). Finally, I think that the text could benefit from both a careful reading by the Authors and an English grammar and spells checking by mother tongue reader or professional English editorial service. Please find below my concerns (Please note that line numbers refer to the *.pdf version of the manuscript).

MAJOR COMMENTS

The Authors in their manuscript analyze the chronologies splitting them in two uneven parts, calculating Pearson’s coefficients between predictor and predictand and valuating them as an index of sensitivity of the species to climate. It is correct but, the fact that an increasing trend is visible only in the two chronologies representative of the high-altitude plots and the fact that, at the same time, they represent the oldest chronologies, lead to think that the increasing trend observable at the end of the chronologies could be a bias induced by the chosen standardization method (Negative Exponential Curves). Thus, I suggest to test if other standardization methods on the same datasets return similar results. Moreover, also the shift from positive to negative correlation values in the low altitude stands (even if the entity of this shift is never reported in the manuscript) seems to be induced by the different long-term trend showed by the TRW data and temperature. Thus, for a more comprehensive study, I suggest to consider also the results of other standardization methods applied at the same datasets and, moreover, would be interesting to know if the observed sensitivity, and related changes, is linked only to the mid-frequencies or if it is related also to the high-frequencies. Regarding the latter, it would be interesting to know if the analyzed species is sensitive to high-frequencies climate variations in the past and if the sensitivity is changed under the pressure of the ongoing climate change at the different altitudes as well as the sensitivity to mid-frequency already analyzed. With this aim, a more attention has to be given to the RES chronologies.

Response: Thank you for pointing it out. The correlations of RES chronologies index and climate data can be seen in Fig 6 and Fig 7.

Page 6 Lines 159–163: the authors state that three different standardization methods were used but only two are reported. Moreover, Melvin and Briffa (2014) state: “In RCS, as normally implemented to date, it is assumed that any influence of the common signal on the shape of the RCS curve will be removed by the process of realigning measurement series by ring age and averaging. In effect, large numbers of sub-fossil (or historical or archaeologically sourced) trees which grew over widely differing periods are needed for this averaging process to be effective (Briffa et al., 1992, 1996).”. The problems associated to the use of a small datasets composed by only modern specimens are also reported in Briffa and Melvin, 2011. The necessity to include subfossils or archaeological samples in the dataset to obtain more robust results from the RCS standardization was highlighted also in (e.g.) Büntgen et al. 2006. The signal-free approach could in part overcomes to this issue but only if the age range of the sample cohort is longer than one quarter the length of the chronology (Melvin and Briffa, 2014). Thus, I personally suggest to apply the signal free approach to the proposed dataset if the authors would consider the RCS results.

Briffa, K.R., Jones, P.D., Bartholin, T.S., Eckstein, D., Schweingruber, F.H., Karlén, W., Zetterberg, P., Eronen, M., 1992. Fennoscandian Summers from AD 500: tem- perature changes on short and long timescales. Climate Dynamics 7, 111–119.

Briffa, K.R., Jones, P.D., Schweingruber, F.H., Karlén, W., Shiyatov, S.G., 1996. Tree- ring variables as proxy-climate indicators: problems with low frequency signals. In: Jones, P.D., Bradley, R.S., Jouzel, J. (Eds.), Climatic Variations and Forcing Mechanisms of the Last 2000 Years. Springer-Verlag, Berlin, pp. 9–41.

Briffa KR and Melvin TM (2011) A closer look at regional curve standardization of tree-ring records: justification of the need, a warning of some pitfalls, and suggested improvements in its application. In Dendroclimatology, pp. 113-145.

Springer, Dordrecht.

Melvin, T.M., Briffa, K.R., 2014. CRUST: Software for the implementation of Regional Chronology Standardisation: Part 1. Signal-Free RCS. Dendrochronologia 32, 7–20. https://doi.org/10.1016/j.dendro.2013.06.002

Moreover, I wonder why Authors decide to present and discuss results of a correlation window that overlaps two shorter windows (i.e. 1927–2016 versus 1927–1969 and 1970–2016). I can hardly find the added value of this analysis to the manuscript except making more confusing and complicated the result section. From my point of view it is obvious that the results obtained by the “1927–2016” period will represent a mixed signal of the fraction of the shorter windows in function to the length of the overlapped period. Moreover, I am still convinced that the notation 1927–* is misleading in the text and that the Authors should find an alternative notation or limit the analysis to the longer period covered by all the chronologies (i.e. 1947–2016).

Response: Thank you for pointing it out. The analysis was limited between 1947 and 2016, which can be seen in Fig 6 and Fig 7.

MINOR COMMENTS

Page 3 Line 76: conifers can be also deciduous (e.g. the Larix gmelinii cited by authors few row above). Please, be clearer.

Response: Thank you very much for pointing it out. This was revised in Page 3 Line 76.

Page 3 Lines 80–82: as pointed-out by Review 2, only Latin name must be in italic. Moreover, “Suk” is an abbreviation and thus must be dotted; “Suk.”

Response: Thank you very much for pointing it out. This was revised in Page 3 Line 80.

Page 4 Lines 89–90: as same as lines 80–82.

Response: Thank you very much for pointing it out. This was revised in Page 4 Line 89-90.

Page 6 Line 140: as same as lines 80–82.

Response: Thank you very much for pointing it out. This was revised in Page 6 Line 140.

Page 6 Line 143: I suggest changing “30 m * 30 m” with the simpler “30x30 m”.

Response: Thank you very much for pointing it out. “30 m * 30 m” has revised as “30x30 m” in Page 6 Line 143.

Page 6 Line 148: the reported number of BP specimens differ from that reported in Table 1. Be consistent.

Response: Thank you very much for pointing it out. This was revised in Page 6 Line 148.

Page 6 Lines 165–166: STD, RES and ARS chronology could be all “the best” chronology thus the best chronology does not exist; the most suitable chronology depends by the aim of the study.

Response: Thank you very much for pointing it out. This was revised in Page 6 Line 164.

Page 7 Lines 167–169: as pointed out in the previous review, it is not clear to me how the EPS could confirm the maximum length of a reliable chronology. Nothing changes from the previous version of the manuscript

despite authors state that they had revised it.

Response: Thank you very much for pointing it out. This was revised in Page 6 Line 165-166.

Page 7 Lines 183–185: the authors state that the climatic data were detrended, but no further information were supplied making the process unreproducible.

Response: Thank you very much for pointing it out. This was revised in Page 7 Line 183-187.

Page 12 Lines 318–320: RBAR and EPS statistics do not represent the climatic information but they only show a certain amount of coherence between the series of a chronology. Please revise the sentence.

Response: Thank you very much for pointing it out. This was revised in Page 12 Line 318-319.

Page 12 Lines 332: it is still reported as “FraxiMus”, again, the correct gender is FraxiNus with an N! Correct it.

Response: Thank you very much for pointing it out. This was revised in Page 12 Line 332

Page 13 Line 335: as same as lines 80–82.

Response: Thank you very much for pointing it out. This was revised in Page 13 Line 335.

Page 13 Line 353: correctly cite the reference.

Response: Thank you very much for pointing it out. The reference was revised in Page 13 Line 352.

Page 13 Line 354: Suk is an abbreviation. Add a full stop to it.

Response: Thank you very much for pointing it out. This was revised in line 353.

Page 13 Line 359: be consistent, use site name or elevation not a mix of them.

Response: Thank you very much for pointing it out. This was revised in Page 13 line 358.

Page 13 Line 360: as same as line 80–82.

Response: Thank you very much for pointing it out. This was revised in Page 13 line 360.

Page 14 Line 381: as same as line 80–82.

Response: Thank you very much for pointing it out. This was revised in Page 14 line 381-382.

Page 14 Line 383–386: the authors state: “Our results showed that at high altitude areas, before 1970 (Figure 6b, Figure 7b) the growth is not sensible to climate change but to climate and the effect of climate change

could be visible through changes in the growth rate or changes in sensitivity of the tree species.”. However, coping a reviewer’s comment (underlined part) and paste it out of the original context does not increment the probability of the manuscript to receive better comments by the same reviewer especially if the resulting sentence lacks of any scientific sense.

Page 15 Line 392: as same as lines 80–82.

Response: Thank you very much for pointing it out. This was revised in Page 15 line 390.

Page 15 Line 408: as same as lines 80–82.

Response: Thank you very much for pointing it out. This was revised in Page 15 line 404.

Page 15 Line 411: as same as lines 80–82.

Response: Thank you very much for pointing it out. This was revised in Page 15 line 407.